# Proenkephalin deletion in hematopoietic cells induces intestinal barrier failure resulting in clinical feature similarities with irritable bowel syndrome in mice

Xavier Mas-Orea[1,5], Lea Rey [1,5], Louise Battut [1], Cyrielle Bories [2], Camille Petitfils[1], Anne Abot[1,4], Nadine Gheziel [1,2], Eve Wemelle[1], Catherine Blanpied[1], Jean-Paul Motta [1], Claude Knauf [1], Frederick Barreau[1], Eric Espinosa [1], Meryem Aloulou [2], Nicolas Cenac [1], Matteo Serino [1], Lionel Mouledous [3], Nicolas Fazilleau [2] & Gilles Dietrich [1✉]

Opioid-dependent immune-mediated analgesic effects have been broadly reported upon inflammation. In preclinical mouse models of intestinal inflammatory diseases, the local release of enkephalins (endogenous opioids) by colitogenic T lymphocytes alleviate inflammation-induced pain by down-modulating gut-innervating nociceptor activation in periphery. In this study, we wondered whether this immune cell-derived enkephalin-mediated regulation of the nociceptor activity also operates under steady state conditions. Here, we show that chimeric mice engrafted with enkephalin-deficient bone marrow cells exhibit not only visceral hypersensitivity but also an increase in both epithelial paracellular and trans-cellular permeability, an alteration of the microbial topography resulting in increased bacteria-epithelium interactions and a higher frequency of IgA-producing plasma cells in Peyer's patches. All these alterations of the intestinal homeostasis are associated with an anxiety-like behavior despite the absence of an overt inflammation as observed in patients with irritable bowel syndrome. Thus, our results show that immune cell-derived enkephalins play a pivotal role in maintaining gut homeostasis and normal behavior in mice. Because a defect in the mucosal opioid system remarkably mimics some major clinical symptoms of the irritable bowel syndrome, its identification might help to stratify subgroups of patients.

[1] IRSD, Université de Toulouse, INSERM, INRAE, ENVT, Univ Toulouse III - Paul Sabatier (UPS), Toulouse, France. [2] INFINITy, Université de Toulouse, INSERM U1291, CNRS U5051, Univ Toulouse III - Paul Sabatier (UPS), Toulouse, France. [3] Research Center on Animal Cognition (CRCA), Center of Integrative Biology (CBI), Université de Toulouse, CNRS UMR-5169, Univ Toulouse III - Paul Sabatier (UPS), Toulouse, France. [4] Present address: Enterosys SAS, Labège, France. [5] These authors contributed equally: Xavier Mas-Orea, Lea Rey. ✉email: gilles.dietrich@inserm.fr

Nociceptors classified as peptidergic and non-peptidergic represent a subset of sensory nerves. Upon activation, peptidergic primary afferent neurons mainly release calcitonin-gene-related peptide and substance P while non-peptidergic nociceptors do not. Both of these two nociceptor subsets innervating mouse colorectum express MOR and DOR in normal conditions[1,2]. Their expression levels in enteric nervous system increase upon colitis[3–6]. A number of studies in both humans and rodents showed that opioids locally released by innate and adaptive immune cells within the inflammatory site regulate pain intensity by acting on opioid receptors expressed on nociceptor cell surface[7–12]. In this context, we have previously shown that, because of their ability to produce enkephalins, Th1, Th17, and Th1/17 lymphocytes responsible for the development of colitis paradoxically dampen inflammation-induced abdominal pain in mice[13–15]. This analgesic property of effector T lymphocytes operates in acute and chronic intestinal inflammatory models[15–17].

In steady state, most of the lymphocytes diffusely scattered throughout the colonic mucosa are effector memory T lymphocytes[18] generated upon sampling of microbiota-derived antigens[19]. Their ability to be restimulated in response to low doses of antigens and lower co-stimulation signals allow them to rapidly prevent the growth of invading microbes. Although, tissue-resident CD8+ T lymphocytes have been extensively studied, effector memory CD4+ and CD8+ T lymphocytes permanently residing within the mucosa are essential to protect colonic tissue from reoccurring or reinfecting microbes[20]. Considering that enkephalin synthesis is up-regulated upon antigen activation[12,21], we hypothesized that in addition to their protective immune activity, memory T lymphocytes may also regulate basal visceral sensitivity. In line with this hypothesis, lymphocyte-deficient (severe combined immunodeficiency syndrome) SCID mice exhibit a higher visceral sensitivity than immunocompetent syngeneic BALB/C mice[22]. This hypersensitivity is normalized 12 weeks following adoptive transfer of T lymphocytes originating from BALB/C mice[22].

In this study, we have examined whether enkephalins produced by immune cells at steady state control spontaneous visceral hypersensitivity as well as intestinal hyperpermeability. Precisely, we show that chimeric mice in which the enkephalin encoding gene penk is deleted in hematopoietic cells display not only visceral hypersensitivity but also a number of other alterations that remarkably mimic irritable bowel syndrome (IBS) including increased intestinal barrier permeability and anxiety-like behavior.

## Results

### Penk-deficiency in immune cells induces neither colon inflammation nor alteration of the transit despite of a slight increase in mucosal CD4+ T lymphocytes.

The contribution of immune cell-derived enkephalins in the endogenous regulation of visceral sensitivity in basal conditions was investigated by using bone marrow chimeric mice (the experimental design is summarized in Supplementary Table 1). Hematopoietic chimeras were generated by reconstituting lethally irradiated C57BL/6 wild-type mice with bone marrow cells originating from either congenic adult proenkephalin-knockout (penk−/−) or littermate wild-type (penk+/+) mice. More than 90% of bone marrow cells of the chimeras originated from the donors as assessed by quantifying enk gene locus DNA in the recipients' bone marrow cells (Supplementary Fig. 1). The deletion of the gene coding proenkephalin in immune cells did not induce colonic tissue injury as assessed by colon length and wall thickness as well as both macroscopic and histological analyses (Fig. 1a). No significant differences in the intestinal transit was observed between penk−/− and penk+/+ chimeric mice (Fig. 1b). In agreement with the absence of any modification in the whole transit time, frequency and consistency of the stools (Fig. 1b), the amplitude of the colonic contractions remained unchanged in penk−/− chimeric mice (Fig. 1c).

The transcriptome of colonic cells from penk+/+ and penk−/− chimeric mice was then established by RNA-seq. Principal component analysis (PCA) of global colonic gene expression profile showed no separation between the mice with respect to their genotype group (penk+/+ and penk−/−) (Fig. 1d), suggesting poor differences in the two group transcriptomes. Accordingly, only five genes out of 26,086 (DEGs: Fold change > 2 and FDR < 0.01) were differentially expressed between the two groups of mice as shown in the volcano plot (Fig. 1e). In agreement with the absence of overt inflammation signs in penk−/− chimeras, the expression levels of both cytokine (except for TNFα) and chemokine mRNAs as well as the production rates of the main pro-inflammatory and pro-resolutive bioactive lipids were similar to that of penk+/+ chimeras (Supplementary Figs. 2, 3).

Colonic CD45+ hematopoietic cells were analyzed by multicolor flow cytometry. Unsupervised dimensionality-reduction using UMAP (uniform manifold approximation and projection) algorithm on flow cytometry data (Supplementary Fig. 4) shows that penk-deficiency resulted in a slight but significant increase in the frequency of CD4+ T lymphocytes associated with a reduction of those of some innate immune cell subsets, including macrophages/dendritic cells, type-1 innate lymphoid cells and mast cells (Fig. 2). The slight increased frequency of mucosal CD4+ T lymphocytes was not associated with any alteration in the frequencies of conventional and regulatory T lymphocytes (Fig. 3a, b). The activity of the three main subsets of effector CD4+ T lymphocytes including IFNγ-producing Th1 lymphocytes, IL-17-producing Th17 lymphocytes and IL13-producing Th2 lymphocytes remained unchanged (Fig. 3c). No modification of the relative content in naive and activated conventional T lymphocytes and Treg was observed between penk+/+ and penk−/− chimeras in spleen (Fig. 3d) or Peyer's patches (Supplementary Fig. 5). Similarly, penk-deficiency did not impact the frequency of activated CD8+ T lymphocytes in the colonic mucosa, the Peyer's patches and the spleen (Supplementary Fig. 6). In order to identify the most potent enkephalin producers within immune cell populations, CD45+CD11b+ myeloid cells, CD45+TCRβ-B220+ B lymphocytes, CD45+TCRβ+CD8+ T lymphocytes and CD45+TCRβ+CD4+ T lymphocytes were isolated from spleen by fluorescence-activated cell sorting. The different cell populations, more than 94% pure, were then assessed for their penk mRNA content by real-time qPCR. Penk mRNA was only detected in CD4+ T lymphocytes. The penk mRNA synthesis was more than 70% reduced in CD4+ T lymphocytes purified from penk−/− chimeras (Fig. 3e).

### Penk-deficiency in immune cells induces visceral hypersensitivity as well as intestinal hyperpermeability associated to an alteration of the spatial organization and composition of the gut microbiota.

The inability of memory CD69+ CD4+ T lymphocytes, virtually all CD4+ T lymphocytes residing within the colon mucosa (Fig. 3a), to produce enkephalins resulted in an increase in the visceral motor response to colorectal distension (Fig. 4a). This hypersensitivity of the nociceptors innervating the gut was associated with an enhancement of the paracellular intestinal permeability in vivo (Fig. 4b). As shown ex vivo, by using biopsies of Peyer's patches, ileum and colon mounted in Ussing chamber, the increase in permeability observed in penk−/− chimeras was not restricted to the colonic area displaying

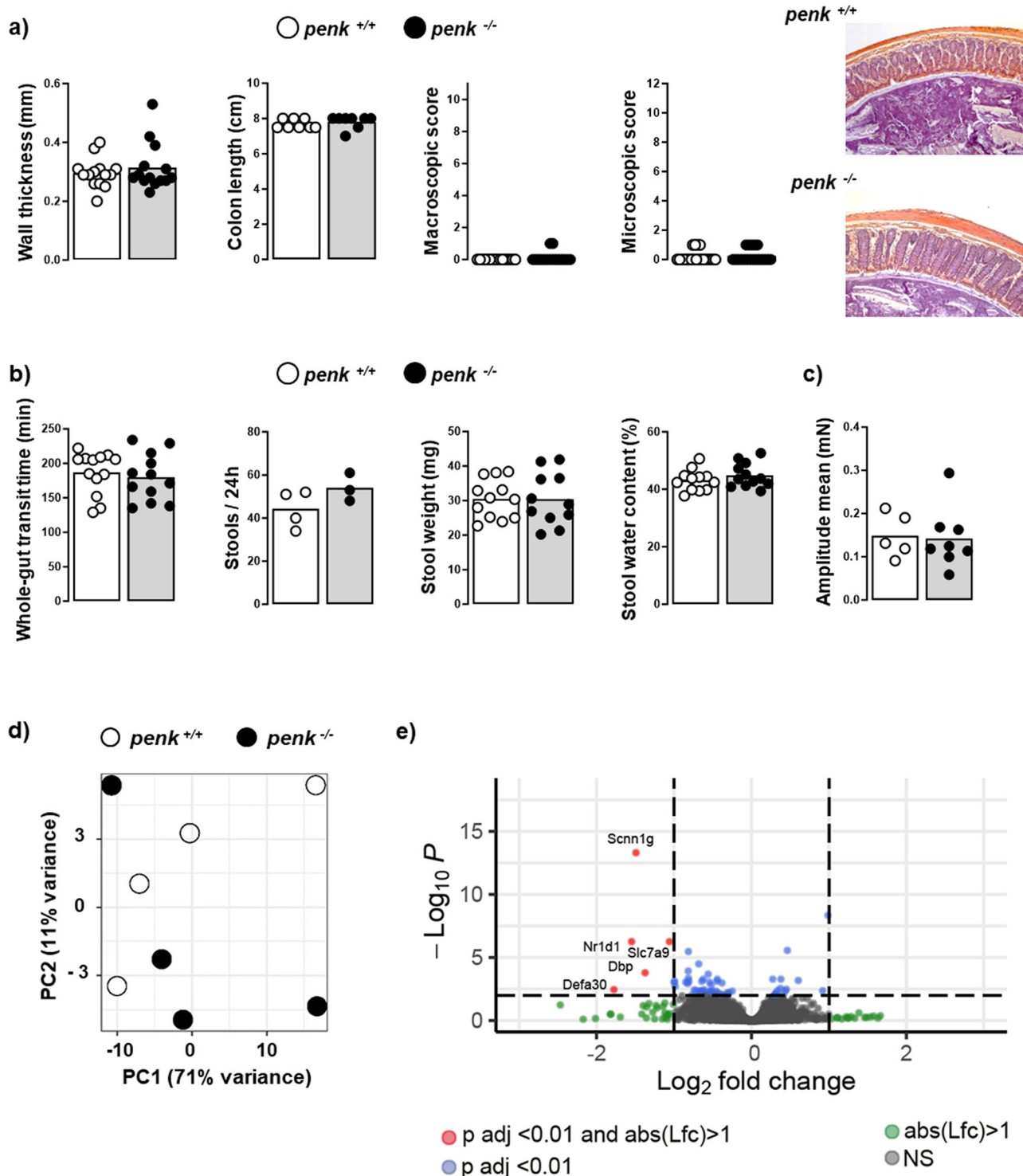

hypersensitivity but was also found in small intestine and, more specifically, in the Peyer's patches mainly involved in luminal antigens sampling (Fig. 4c). In vivo experiments using high molecular weight dextran indicated that the permeability of the epithelial barrier was also increased for macromolecules (up to 40 kDa) (Fig. 4d). This increased passage of the macromolecules across the epithelium was mostly due to an increased transcellular permeability since epithelial cell apoptosis rates were similar between the two groups of mice (Fig. 4e).

The colonic epithelium is covered by a dense mucus layer ('inner layer') free of bacteria and, above it, a loosely adherent

mucus layer ("outer layer") housing most of the bacteria living in dense communities. In healthy conditions, local adherent microbial communities reside on the surface of the inner mucus layer. Because, in a number of pathological conditions including low-grade inflammation, the density and/or the morphotypes of bacteria as well as their ability to interact with epithelium or to translocate into lamina propria may change[23], we investigated the spatial organization of the colonic bacterial flora. Colon microbiota analysis showed that penk-deficiency in mucosal immune cells altered spatial organization of luminal bacteria, scored as biofilm damage (Fig. 5a). Mucosal penk-deficiency was

**Fig. 1 *Penk*-deficient chimeric mice exhibit neither colon inflammation nor gut transit alteration.** Lethally irradiated C57BL/6JRj wild-type mice were engrafted with bone marrow cells from either *penk*$^{+/+}$ (white circles and white histogram) or *penk*$^{-/-}$ (black circles and gray histogram) mice. Eighteen weeks after, chimeric mice were examined for inflammation-induced colonic injury (**a**), bowel transit (**b**, **c**) and global gene expression including inflammatory parameters by bulk RNA sequencing (**d**, **e**). **a** Colonic wall thickness ($n = 14$), colonic length ($n = 8$), macroscopic ($n = 14$) and histological tissue damage ($n = 14$). A representative histopathological analysis performed on H&E-stained colon section shows normal epithelial architecture with neither cellular infiltration nor edema in both *penk*$^{+/+}$ and *penk*$^{-/-}$ chimeric mice (**a**, right panels). **b** Bowel habits including gut transit time, frequency and consistency of stools ($n = 11$ to 13). **c** Colon contractions ($n = 5$ *versus* 8). Each symbol represents one mouse. Statistical analysis was performed using Mann-Whitney U test. **d** Principal component analysis (PCA) performed between transcriptional profiles of colonic cells from *penk*$^{+/+}$ and *penk*$^{-/-}$ chimeric mice. PCA plot of DESeq2 rlog-normalized RNA-seq data. The percent of variance explained by each component is reported in parentheses. Samples are shown in the 2D plane, spanned by their first two principal components. Each mouse is plotted as an individual data point ($n = 4$). **e** Volcano plot depicting differential gene expression in colonic cells from *penk*$^{-/-}$ and *penk*$^{+/+}$ chimeras. Genes with | log$_2$(fold change)| > 1 and FDR < 0.01 (significantly differentially expressed genes) are in red, genes with | log$_2$(fold change)| > 1 but FDR ≥ 0.01 are in green, and genes with | log$_2$(fold change)| ≤ 1 but FDR < 0.01 are in blue, and the rest are in gray.

associated with a bacterial invasion of the mucus layer and an increase in the number of epithelium-bacteria contacts (Fig. 5b) although the mucus thickness remained unchanged (Fig. 5c). These alterations did not significantly impact both the general profile of gut microbiota (Fig. 5d) and several diversity indices (Fig. 5e). A specific gut microbial signature could, however, be identified in *penk*$^{-/-}$ chimeric mice, including a higher abundance for species such as *Lactobacillus reuteri* and *Odoribacter splanchnicus*, together with other taxa such as *Bacilli, Bacteroidetes* and *Prevotella* (Fig. 5f).

IgA antibodies produced by plasma cells generated in gut-associated lymphoid tissues through a non-inflammatory antigen sampling process play a pivotal role in protection against microbial invasion. Peyer's patches mostly in the ileum of the small intestine represent the primary site of IgA-producing plasma cells generated in response to the intestinal bacteria or their products through a T cell-dependent mechanism. Considering T cell-independent IgA responses to commensal bacteria within isolated lymphoid follicles scattered all along the intestine as alternative, we focused on Peyer's patches. In line with an intestinal transcellular hyperpermeability, the frequency of IgA-producing plasma cells was enhanced in Peyer's patches from *penk*$^{-/-}$ chimeric mice (Fig. 6a, b). Accordingly, the frequency of activated B lymphocytes remaining within germinal centers was reduced (Fig. 6b). The strengthening of the adaptive B cell response was testified by the slight but significant increased frequency of memory B cells in the spleen (Fig. 6c). The production of IgA was however not enough to significantly increase the relative number of IgA-bound bacteria in the feces (Fig. 6d).

***Penk*-deficiency in immune cells induces anxiety-like behavior**. To examine whether the lack of immune cell-derived enkephalins induced behavioral changes, we used a number of tests recapitulating various phenotypes of anxiety and depression. The anxiety-like behavior of the mice was first assessed using open field (measurement of locomotor and exploratory activity) and elevated plus maze (based on the conflict between exploration of a novel environment and the natural aversion of rodents for heights and open spaces) 1-day apart. After two resting days, mice were assessed for depressive-like behavior using tail suspension test (development of an immobile posture rather an escape-related behavior in response to stress induced by the tail suspension) and then novelty-suppressed feeding after two additional resting days. Open field, elevated plus maze and tail suspension did not reveal any behavioral differences between *penk*$^{+/+}$ and *penk*$^{-/-}$ chimeric mice (Supplementary Fig. 7). The anxiety-like behavior of the mice with *penk*-deficient immune cells was only unveiled in the novelty-suppressed feeding test. This test is based on hyponeophagia phenomenon in which innate fear of mice to a novel environment counterbalances hunger-induced behavior. Animals were deprived for food 24 h prior the behavioral test. The loss of weight resulting from food deprivation was similar between the two groups of mice (Fig. 7a) suggesting that *penk*$^{-/-}$ and *penk*$^{+/+}$ chimeric mice should have a similar motivation to feed during the test. As shown in Fig. 7b, the time up to *penk*$^{-/-}$ chimeras express a feeding behavior in an anxiogenic environment was significantly higher than that of *penk*$^{+/+}$ chimeras highlighting an anxious-like behavior. The absence of any difference in the feeding behavior between the two groups of animals was confirmed by the consumption of food measured at the end of the test in the home cage (Fig. 7c). The anxiety-like behavior of *penk*$^{-/-}$ chimeras was probably revealed in the novelty suppressed feeding test because of the higher anxiogenic properties of this test compared to the open field and elevated plus maze setups.

**Discussion**

It is now well established that innate immune cells such as neutrophils[11], monocytes/macrophages[7] or dendritic cells[21] as well as adaptive immune cells such as CD4$^+$ and CD8$^+$ T lymphocytes[24,25] produce opioids. In infection-mediated inflammatory response, CD4$^+$ T lymphocytes, that appear to be the main producers of enkephalins of immune origin, play a pivotal role in the endogenous regulation of inflammatory pain in mice[21,26,27]. Enkephalin-mediated analgesic properties of effector CD4$^+$ T lymphocytes has been reported in a number of mouse models of intestinal inflammation including DSS treatment[15,17,28,29], CD4$^+$ CD45RB$^{high}$ T cell transfer[15] or IL10-deficiency[16]. In these models, colitogenic CD4$^+$ T lymphocytes generated in response to intestinal microbiota alleviate abdominal pain through the local release of enkephalins which synthesis is up-regulated upon antigen-priming[21]. Because under normal conditions, colonic mucosa from healthy individuals contain resident memory T lymphocytes generated upon gut microbiota sampling[30], we hypothesized that the mucosal opioid tone dependent on colonic resident effector T lymphocytes may affect basal visceral sensitivity and, as a consequence, intestinal homeostasis. Indeed, effector CD4$^+$ T cells generated within Peyer's patches, isolated lymphoid follicles, and mesenteric lymph nodes in response to microbiota-derived antigens populate intestinal mucosa to stay for very long time. These so-called tissue-resident memory T lymphocytes, downregulated by the regulatory T lymphocytes, are able to secrete rapidly cytokines upon a new contact with cognate antigen. Because enkephalin release is also dependent on local re-stimulation of effector memory T lymphocytes by antigen[27], it is most likely that enkephalins are secreted together with cytokines upon physiological microbiota-derived antigen sampling. In agreement, we found that the vast majority of the CD4$^+$ and CD8$^+$ T lymphocytes within the colon *lamina propria* expressed the activation and/or resident memory

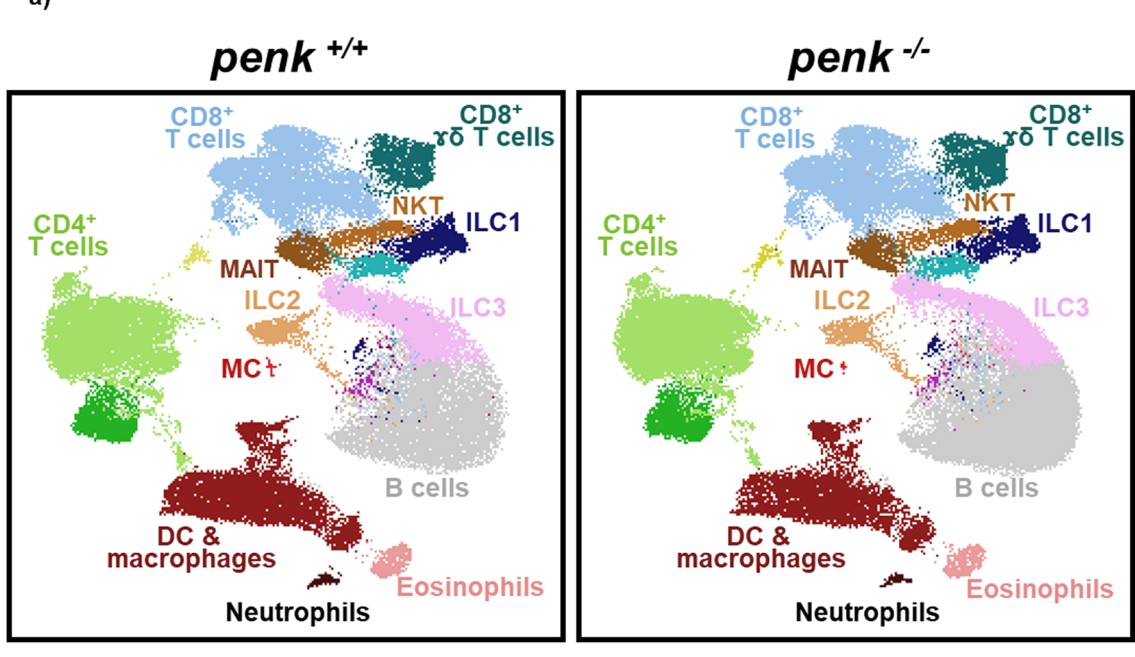

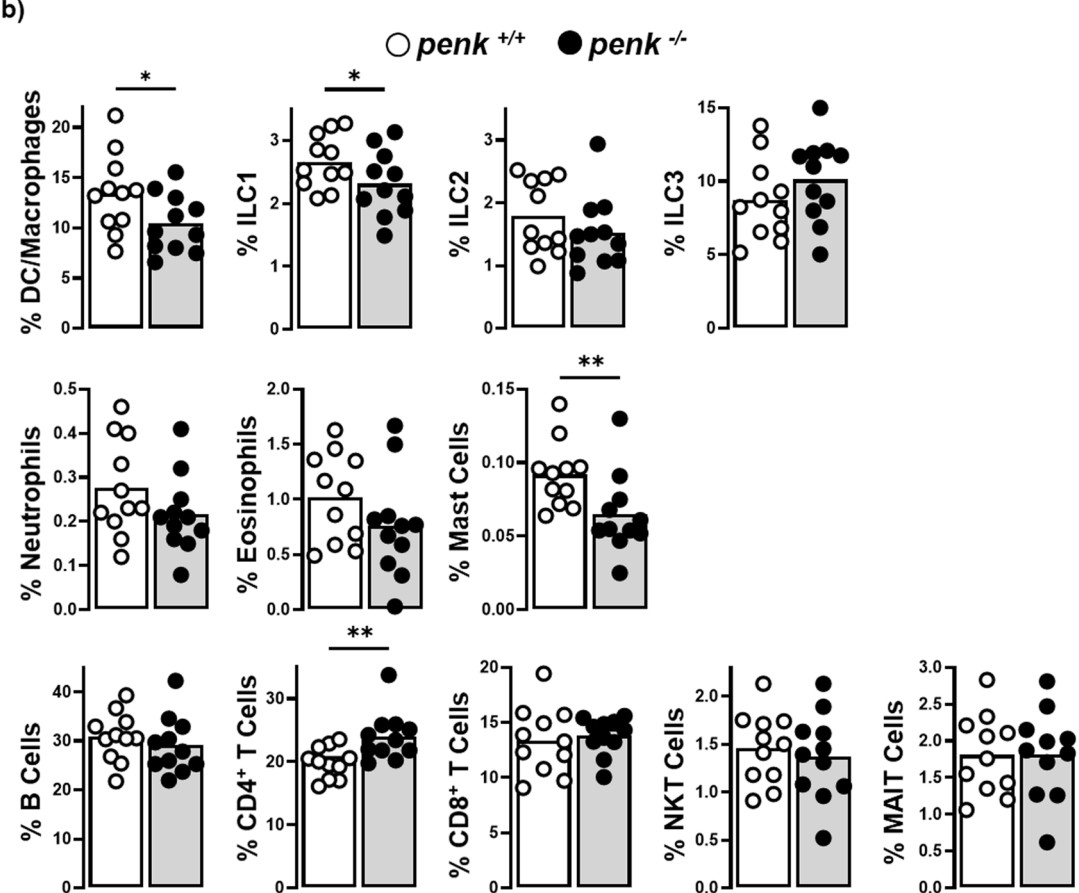

**Fig. 2 *Penk*-deficient chimeric mice exhibit alteration in the relative frequency of mucosal immune cells.** Cytofluorometric analysis of the relative frequency of innate and adaptive immune cells in colon mucosa of *penk*$^{+/+}$ (white circles and white histogram) and *penk*$^{-/-}$ (black circles and gray histogram) chimeric mice (*n* = 11). **a** Deep phenotyping of CD45$^+$ cells performed using Phenograph for clustering and UMAP for dimensionality reduction. Phenograph clusters are annotated according to characteristic marker expression (Supplementary Fig. 4). Nearest-neighbors clusters were unsupervised delineated using phenograph plug-in. **b** Frequencies of mucosal immune cell subsets identified by this method. Each symbol represents one mouse. Statistical analysis was performed using Mann-Whitney U test. * *p* < 0.05; ** *p* < 0.01.

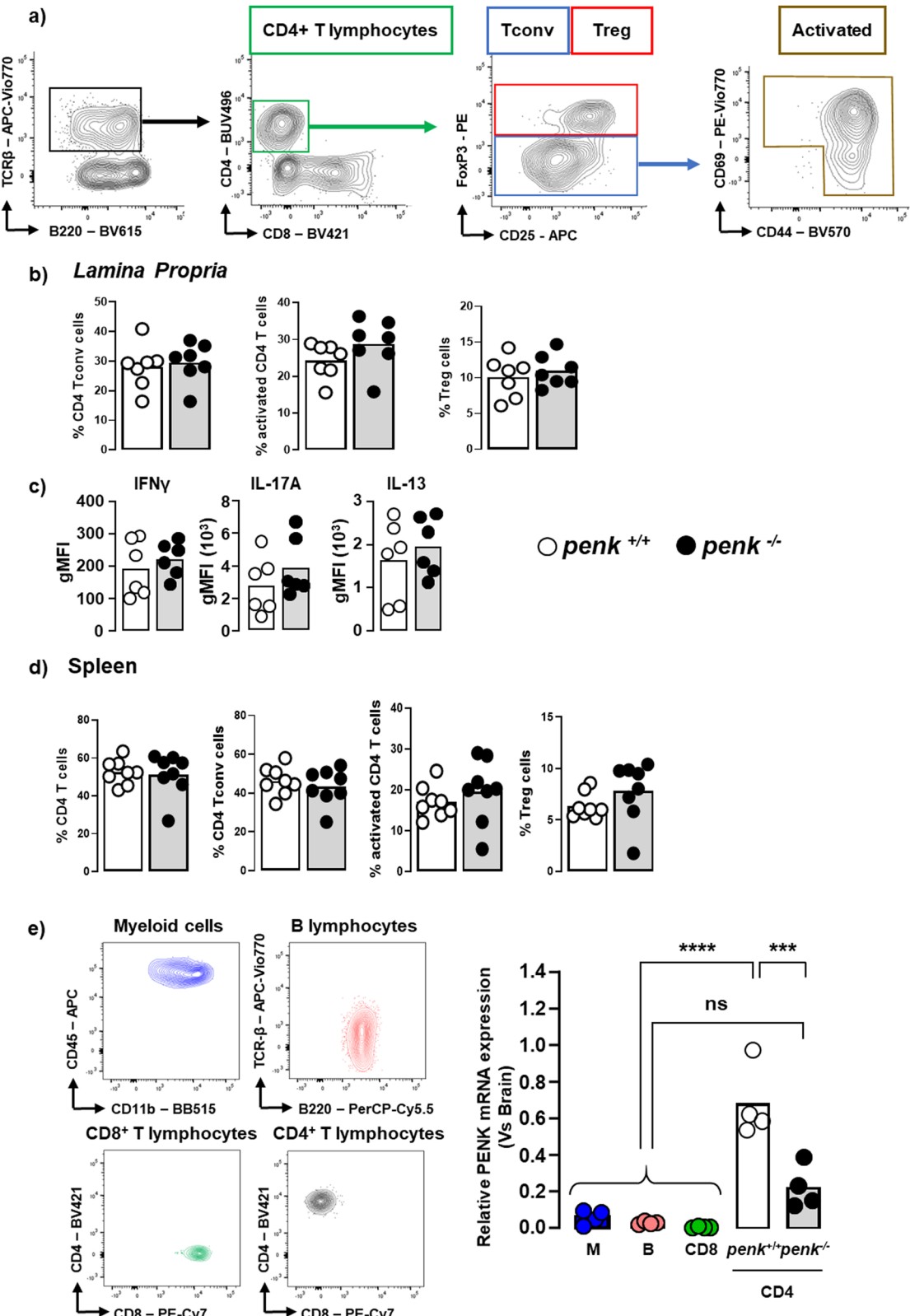

marker CD69 and only CD4+ T lymphocytes including regulatory T cells, but not CD8+ T cells, expressed *penk* mRNA.

When CD4+ T cells do not produce enkephalins, *penk*−/− mice display visceral hypersensitivity suggesting that bacterial products from gut lumen and/or physiological inflammation resulting from mucosal immune surveillance constitutively stimulate sensory neurons[31–33]. In normal conditions, this basal

neuronal activation would be down-regulated by mucosal CD4+ T cell-derived opioid tone. This basal mucosal opioid tone is, however, insufficient when massive bacterial invasion occurs following epithelial barrier injury as in the acute phase of DSS-induced colitis[12,17]. The reduction of the enkephalin tone in basal situations also resulted in an increase in both epithelial paracellular and transcellular permeability together with a spatial

**Fig. 3 *Penk*-deficient chimeric mice display virtually no functional T cell alteration.** The relative frequency of CD4[+] T lymphocyte subsets in colonic *lamina propria* and spleen from *penk*[+/+] (white circles and white histogram) and *penk*[−/−] (black circles and gray histogram) chimeric mice was estimated by cytofluorometry. **a** Gating strategy for flow cytometry data analysis. Living CD45-expressing cells gated on TCRβ chain[+] CD4[+] T lymphocytes were immunophenotyped as FoxP3[+] regulatory T cells and FoxP3[-] conventional T lymphocytes. Activated conventional CD4[+] T lymphocytes were then distinguished based on the expression of CD44 and/or CD69 cell surface markers. **b** Frequency of conventional CD4[+] T lymphocytes, activated conventional CD4[+] T lymphocytes and regulatory T cells among living CD45[+] TCRβ-expressing cells within the *lamina propria* (n = 7). **c** IFNγ (Th1), IL-17 (Th17), and IL-13 (Th2) cytokine expression levels expressed as geometric mean of fluorescence intensity (gMFI) in CD4[+] T lymphocytes from colonic *lamina propria* (n = 6). **d** Frequency of CD4[+] T lymphocytes, conventional CD4[+] T lymphocytes, activated conventional CD4[+] T lymphocytes and regulatory T cells among living CD45[+] TCRβ-expressing cells within the spleen (n = 8). Each symbol represents one mouse. Statistical analysis was performed using Mann-Whitney U test. **e** Spleen cells from chimeric *penk*[+/+] and *penk*[−/−] mice (n = 4) were isolated by cell-sorting as myeloid cells, B lymphocytes and CD4 and CD8 T lymphocytes based on the expression of the cell surface antigens CD45/CD11b, B220, TCRβ chain/CD4 and TCRβ chain/CD8 respectively (left panels). *Penk* mRNA expression was quantified by real-time PCR. mRNA content was normalized to the *HPRT* mRNA and quantified relative to standard mouse brain cDNA (n = 4). Each symbol represents one mouse. Statistical analysis was performed using one-way ANOVA followed by Sidak's multiple comparison test; ***p < 0.001, ****p < 0.0001.

redistribution of the luminal bacteria. In chimeras with *penk*[+/+] hematopoietic cells, microbiota within the intestinal lumen mostly does not interact with host cells. By contrast, in *penk*-deficient chimeric mice, bacteria infiltrate the mucus until the vicinity of the epithelial lining although the mucus thickness was not decreased, as already reported at the onset of the DSS-induced colitis and in IL-10[−/−] mice with low grade of inflammation[34]. Reduced levels of mRNA coding key proteins (*ECP, S100A7, Ltf, and PIgR*) (Supplementary Fig. 8) in the protection of colonic mucosa to microbial challenges are in line with the presence of bacteria closer to the epithelial cells in *penk*-deficient chimeras. Although the bacterial diversity remained unchanged, some mucin-degrading bacterial members, including *Prevotella spp* and *Odoribacter splanchnicus*[35,36] were increased. These two latter bacterial species have been reported differentially abundant in IBS patients and healthy controls[37,38]. Bacteria relocalized close to the epithelium, a clinical feature also found in almost two-thirds of IBS patients[39], might then locally trigger immune response via bacterial metabolites, bacterial components, or bacteria themselves. The increased frequency of both mucosal CD4[+] T lymphocytes and IgA-producing plasma cells in the Peyer's patches testify that an immune response takes place to limit commensal bacteria invasion. Since *penk*[−/−] hematopoietic chimeras do not develop spontaneously colitis, it could be assumed that, at least in the few weeks following engraftment of *penk*[−/−] immune cells, the local immune response to bacteria is sufficient to protect animals from infection[40]. Indeed, intestinal permeability defect does not always result in intestinal inflammation as it could be observed in some healthy first-degree relatives of patients with inflammatory bowel diseases such as parents, children, or siblings[41,42]. Similarly, although some of patients with IBS have an increased permeability, they do not exhibit obvious gut injury[43,44] as it is described in inflammatory bowel diseases[43,44]. In line with studies reporting an association between intestinal permeability failure, visceral hypersensitivity and psychological disorders[45,46], *penk*-deficient chimeric mice display anxiety-like behavior, a psychological comorbidity frequently associated with IBS[47].

Taken together, our results show that mice in which the mucosal opioid tone is dramatically reduced because of the inability of immune cells to produce enkephalins recapitulate a number of symptoms for IBS diagnosis including intestinal hyperpermeability, visceral hypersensitivity, anxiety, slight alterations of microbiota composition and subtle immune activation[48]. In line with a potential loss of endogenous regulation of colonic sensory afferents by immune-derived opioids in IBS, β-endorphin expression has been found reduced in colonic biopsies from IBS patients as compared to healthy controls[49,50]. However, contrasting with IBS patients, both the intestinal transit and the

colonic contractility remained unchanged in *penk*-deficient mice, maybe because of the absence of T cells within the myenteric plexus in normal conditions[51]. Of note, inflammation status worsening over time may result in bowel dysfunction. Because irradiation induces intestinal epithelium injury[52,53], it could also be hypothesized that a reduced opioid tone delays mucosal healing[54]. A hypothesis supported by the significant reduction in the mRNA expression levels of a number of genes such as *Snai1*, *Fn1*, *Egf*, *Atg16l*, and *AhR* involved in intestinal epithelium regeneration (Supplementary Fig. 8). Moreover, chronic treatment of normal non-irradiated mice with naloxone methiodide (an antagonist of all the three opioid receptor classes that does not cross the blood-brain barrier) does not alter intestinal permeability (Supplementary Fig. 9). It remains that lacking of endogenous immune cell-derived opioid tone maintains hyperpermeability and, thereby could increase the probability of developing IBS.

IBS is not a life-threatening disease but chronic pain and psychiatric comorbidity are at the origin of a worsening of patient's quality of life often more severe than that observed in other chronic intestinal diseases such as Crohn's disease[55]. The absence of knowledge about the etiology as well as the different pathophysiological mechanisms that may operate in the various subtypes of patients[56] leads to frequent therapeutic failure[57]. In this context, our results, suggesting that a defect in mucosal immune cell-derived opioid signaling could be involved in the pathophysiology of IBS and used to define an IBS subgroup of patients, might unravel original therapeutic opportunities[58–60].

## Materials and methods

**Generation of hematopoietic *penk*[−/−] or *penk*[+/+] chimeric mice.** Bone marrow cells originating either from 9–11-week-old male preproenkephalin-knockout (*penk*[−/−]) mice (B6.129-Penk-rstm1Pig/J strain, the Jackson Laboratory, Bar Harbor, Maine, USA) or from age-matched male wild-type (*penk*[+/+]) mice were injected into 7-week-old male C57BL/6JRj recipient mice lethally irradiated (1100 rad). The mice were housed in groups of 4 or 5 in ventilated cages with chow and water ad libitum for 18–21 weeks before to be examined. All procedures involving animals were performed in accordance with the Guide for the Care and Use of Laboratory Animals of the European Council and were approved by the Animal Care and Ethics Committee of US006/CREFRE (CEEA-122; application number APAFIS #16385-2018080222083660 v3). We have complied with all relevant ethical regulations for animal use.

Chimerism was assessed by quantifying *enk* gene locus DNA relative to *enk* gene locus plus *neomycin* gene in the recipient's bone marrow cells. Genomic DNA was extract from bone marrow cells depleted in red blood cells from chimeras using kit Extract-

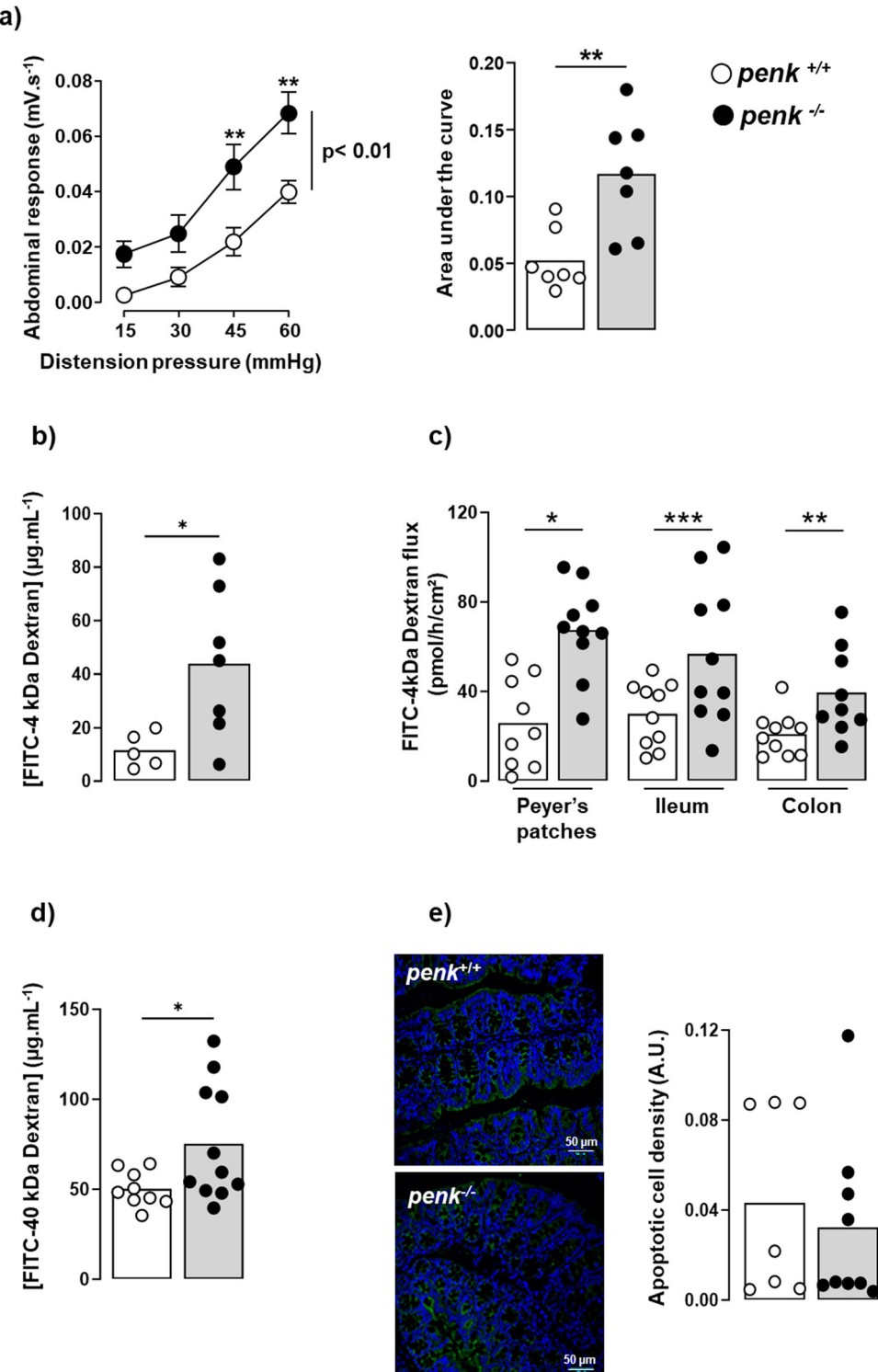

N-Amp™ Tissue PCR Kit (Sigma-Aldrich, St Louis, MO, USA). *Enk* gene locus (*penk*$^{+/+}$) and Neomycin gene (*penk*$^{-/-}$) were quantified by PCR using respectively the following forward and reverse primers 5'- TGC CTT GAA TGG CTT TCT CT- 3' (oIMR1861) / 5'- GTT GTC TCC CGT TCC CAG TA - 3' (oIMR1862) and 5'- CTT GGG TGG AGA GGC TAT TC - 3' (oIMR6916) / 5'- AGG TGA GAT GAC AGG AGA TC - 3' (oIMR6917) (JAX, Sacramento, CA, USA). PCR amplicons were run in 2% agarose gel, stained with GelRed® Nucleic Acid Gel Stain (biotium, Fremont, CA, USA) for 20 min and then revealed with ChemiDoc XRS$^+$ (Bio-Rad, Hercules, CA, USA). The

fluorescence intensity of the bands corresponding to the *enk* gene locus (*penk*$^{+/+}$) and to the neomycin gene (*penk*$^{-/-}$) was quantified by using Fiji/ ImageJ software (Supplementary Fig. 1).

**Macroscopic assessment of colon**. Macroscopic colonic tissue damage was evaluated using a scale ranging from 0 to 11 as follows: erythema (absent (0), length of the area less than 1 cm (1), more than 1 cm (2)), edema (absent (0), mild (1), severe (2)), strictures (absent (0), one (1), two (2), more than two (3)), ulceration (absent (0), present (1)), mucus (present (0), absent

**Fig. 4 *Penk*-deficient chimeric mice exhibit an increase in both visceral sensitivity and paracellular and transcellular intestinal permeability.** *penk*[+/+] (white circles and white histogram) and *penk*[−/−] (black circles and gray histogram) chimeric mice were assessed for visceral sensitivity and both paracellular and transcellular intestinal permeability. **a** Visceromotor responses to colorectal distension pressures ranging from 15 to 60 mmHg (*n* = 7). Data are expressed as mean ± SEM. Statistical analysis was performed using repeated-measures two-way ANOVA and Sidak's multiple comparison test (right panel). Area under the curve (AUC) calculated by plotting individual visceromotor response is shown in the left panel. Each symbol represents one mouse. Statistical analysis was performed using Mann-Whitney U test. ** *p* < 0.01. **b** In vivo measurement of the paracellular permeability expressed as serum 4 kDa FITC-dextran concentration determined 4 h after gavage (*n* = 5 versus 7). **c** Ex vivo measurement of the paracellular permeability of biopsies from Peyer's patches, ileum and colon mounted in Ussing chamber (*n* = 9 or 10). Intestinal paracellular permeability was expressed as luminal-to-serosal flux of FITC-dextran calculated upon 1-h incubation time. **d** In vivo measurement of the transcellular permeability expressed as serum 40 kDa FITC-dextran concentration determined 4 h after gavage (*n* = 9 versus 11). **e** Representative staining of apoptotic epithelial cells by anti-activated caspase 3 antibodies (left panel) and their quantification (right panel) (*n* = 7 versus 9). Each symbol represents one mouse. Statistical analysis was performed using Mann-Whitney U test. * *p* < 0.05; ** *p* < 0.01; *** *p* < 0.001.

(1)), and adhesion (absent (0), moderate (1), severe (2)). Bowel wall thickness was measured with an electronic calliper in the distal part of the colon, at 0.5 cm above the anus[14].

**Histological assessment of colon tissue.** Colonic biopsies excised at 2 cm from the anus were immediately transferred into 4% paraformaldehyde and embedded in paraffin. Colonic sections (5 μm) were stained with hematoxylin-eosin (H&E) and scored on a scale ranging from 0 to 12. Inflammatory cell infiltration, epithelial/mucosal alteration (including vasculitis, goblet cell depletion and crypt abscesses), mucosal architecture alteration (including ulceration and crypt loss), and submucosal edema were graded from 0 to 3 (absent, mild, moderate and severe)[14].

**Frequency, weight, and water content of stools.** Mouse droppings excreted for 24 h were enumerated, collected and weighted using a chemical balance. Stools were then heat for 48 h at 50 °C to remove water. The stool water content was defined as the ratio of dry weight to wet weight.

**Gut transit time assessment.** Mice were gavaged with 0.5% methylcellulose solution containing 6% Carmine (both reagents were from Sigma-Aldrich). Whole gut transit time was defined as the time to recover Carmine red dye in the fecal stream[61].

**Assessment of colon contractions ex vivo.** Colons were quickly dissected and washed in Krebs–Ringer bicarbonate/glucose buffer (pH 7.4) in 95% $O_2$ and 5% $CO_2$. Colon segments were incubated in 25 mL of oxygenated Krebs-Ringer solution for 30 min at 37 °C and then attached to an isotonic transducer (MLT7006 Isotonic Transducer, Hugo Basile, Comerio, Italy). A 2 g (20 mNewton) load was applied to the lever. Isotonic contractions were recorded using LabChart software (ADInstruments, Inc., Colorado Springs, CO, USA)[62].

**Bulk RNA-sequencing (RNA-Seq).** Transcriptome profiling was performed on total RNAs (extracted as described below) of colonic biopsies originating from four animals in each group of *penk*[+/+] and *penk*[−/−] chimeric mice. The corresponding RNA-seq paired-end libraries were prepared according to Illumina's protocol with some adjustments, using the TruSeq Stranded mRNA library prep Kit (Illumina, San Diego, USA). Briefly, mRNAs were first selected from 10 μg total RNA using poly-T beads. Then, RNAs were fragmented during 2 min and retro-transcribed to generate double stranded cDNA. Compatible adapters were ligated, allowing the barcoding of the samples with unique dual indices. Twelve cycles of PCR were applied to amplify libraries, and a final purification step allowed to obtain 280–1000 pb fragments. Libraries quality was assessed using the HS NGS kit on the Fragment Analyzer (Agilent Technologies,

Santa Clara, USA). Libraries quantification and sequencing were performed at the GeT-PlaGe core facility (INRAe, Toulouse, France). Libraries were quantified by qPCR using the KAPA Library Quantification Kit (Roche, Basel, Switzerland) to obtain an accurate quantification.

The quality control of reads was performed with FastQC and all samples pass classical quality control. Reads were then aligned and quantified with STAR (Version 2.7.8a) on the *Mus musculus* genome GRCm39.107 from EBI databases. Differential analyses of RNA-Seq data at the gene level were performed using the DESeq2 R package (v1.30.0) with the recommended workflow[63].

**Real-time quantitative PCR analysis.** Frozen colon biopsies were crushed in 500 μL TRIzol™ Reagent (Molecular Research Center, Euromedex, Souffelweyersheim, France) in Precellys lysing kit D-tubes (Bertin Technologies, Montigny le Bretonneux, France) placed in a Precellys (6000 rpm, 30 s twice; Bertin Technologies). Total RNA was then isolated by using GenElute™ mammalian total RNA miniprep kit (Sigma-Aldrich) following the manufacturer recommendations and evaluated using a ND-1000 Nanodrop spectrophotometer and gel electrophoresis. Total RNA was reverse-transcribed with Moloney murine leukemia virus reverse transcriptase (Fisher Scientific) using random hexamers (MP Biomedical, Fisher scientific SAS, Illkirch, France) for priming. Transcripts were quantified by real-time PCR on a LightCycler480II (Roche Diagnostics, Meylan, France) using specific forward and reverse primers (Supplementary table 2). The target gene expression was normalized to that of the hypoxanthine-guanine phosphoribosyltransferase. The $2^{-\Delta\Delta CT}$ method was used to evaluate mRNA expression levels in chimeric *penk*[−/−] mice relative to the chimeric *penk*[+/+] control mice[17].

**Quantification of polyunsaturated fatty acids (PUFA) and their metabolites in colonic biopsies.** Polyunsaturated fatty acid (PUFA) metabolites were quantified by mass spectrometry after lipid extraction as previously described[64]. After addition of 200 μL of PBS and 5 μL deuterated Internal Standard mixture (5-HETEd8, LxA4d4, and LtB4d4), colonic biopsies were crushed in lysing MatrixA tubes with a precellys (Bertin Technologies). After two crush cycles (6.5 m s⁻¹, 30 s), 10 μL of suspensions were withdrawn for protein quantification and 0.3 mL of cold methanol (MeOH) were added. Samples were centrifuged at 1016 × *g* for 15 min (4 °C) and the resulting supernatants were submitted to solid phase extraction of lipids using HLB plate (OASIS® HLB mg, 96-well plate, Waters, Saint-Quentin-en-Yvelines, France). Briefly, plates were conditioned with 500 μL MeOH and 500 μL $H_2O$/MeOH (90:10, v/v). Samples were loaded at a flow rate of about one drop per 2 s, and, after complete loading, columns were washed with 500 μL $H_2O$/MeOH (90:10, v/v). The phase was thereafter dried under aspiration and lipids were eluted with

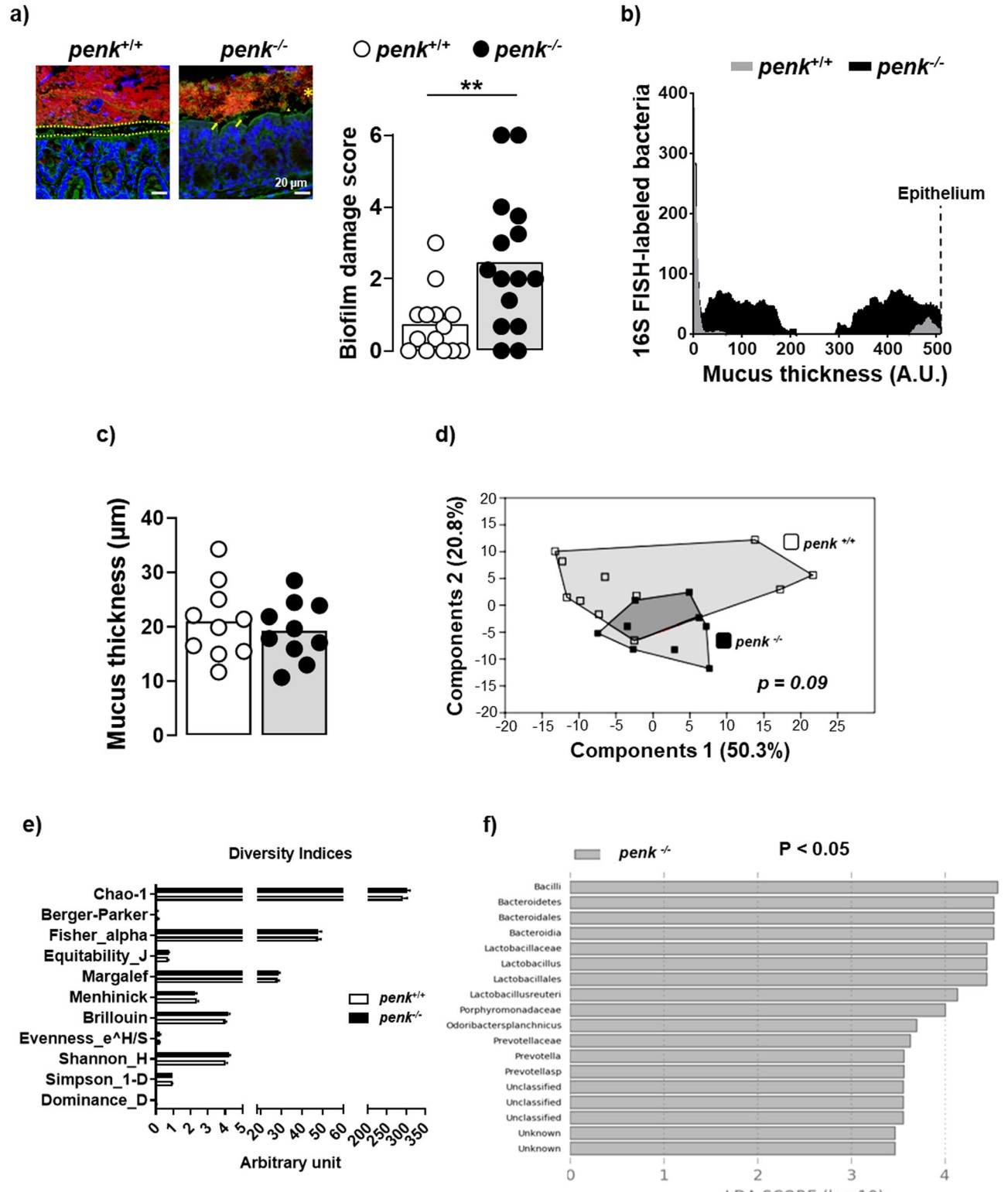

750 µL MeOH. Solvent was evaporated under N$_2$ and samples were resuspended with 140 µL MeOH and transferred into a vial (Macherey-Nagel, Hoerdt, France). Finally, the 140 µL of MeOH were evaporated and our sample resuspended with 10 µL of MeOH for liquid chromatography/mass spectrometry analysis.

6-keto-prostaglandin F1 alpha (6kPGF$_{1\alpha}$), thromboxane B2 (TxB$_2$), PGE$_2$, PGD$_2$, PGF2α, lipoxin A4 (LxA$_4$), resolvin D5 (RvD5), protectin Dx (PDx) were quantified on an ultrahigh-

performance liquid chromatography system (UHPLC; Agilent LC1290 Infinity) coupled to an Agilent 6460 triple quadrupole MS (Agilent Technologies) equipped with electrospray ionization operating in negative mode. Reverse-phase UHPLC was performed using a ZorBAX SB-C18 column (Agilent Technologies) with a gradient elution. The mobile phases consisted of water, acetonitrile (ACN), and formic acid (FA) [75:25:0.1 (v/v/v)] (solution A) and ACN and FA [100:0.1 (v/v)] (solution B).

**Fig. 5 *Penk*-deficient chimeric mice display an altered microbiota spatial organization.** *penk*[+/+] (white circles and white histogram) and *penk*[−/−] (black circles and gray histogram) chimeric mice were assessed for both spatial organization and profile of microbiota. Carnoy's fixed distal colon sections were hybridized with universal probe Eub338 to label bacteria (red color) and counterstained with fluorescein-labeled wheat germ agglutinin and 4′,6-diamidino-2-phenylindole DAPI to highlight polysaccharides-rich mucus layer (green color) and host cell nuclei (blue color) respectively. **a** Representative images show a 20 µm sterile mucus layer separating the epithelium from a dense microbiota biofilm in *penk*[+/+] chimeric mice (left image) and a closer distance of microbiota biofilm to mucosa, mucus layer invasion and visible contacts of bacteria with epithelium in *penk*[−/−] chimeric mice (right image). Scale bars represent 20 µm. The scoring of microbiota biofilm alterations in each group of mice is depicted in the right panel. Each circle represents for one mouse, the mean of blind acquisition of 3–5 different fields ($n = 14$ versus 15). Statistical analysis was performed using Mann-Whitney U test. ** $p < 0.01$. **b** Bacteria penetration into the mucus measured by image processing on Fiji. Pixels corresponding to labeled 16S RNA were enumerated between the edge of the lumen to the edge of the epithelium. The mucus area was manually traced, and an arbitrary distance was assigned to each pixel from the middle of the mucus layer to the edge of the lumen (apical) and to the epithelial cell border (basal). **c** Thickness of the mucus layer from the surface of the epithelial cells to the edge of the luminal biofilm. Each circle represents the mean of blind acquisition of 15 different fields for one mouse ($n = 10$). Statistical analysis was performed using Mann-Whitney U test. **d** Gut microbiota profile estimated using Euclidean distance-based Principal Component Analysis (PCA) (one-way perMANOVA analysis plus Bonferroni correction), (**e**) Diversity indices, (**f**) Linear Discriminant Analysis (LDA) score (Kruskal-Wallis test and threshold on the logarithmic LDA score for discriminative features set at 2) ($n = 11$ versus 9).

The linear gradient was as follows: 0% solution B at 0 minute, 85% solution B at 8.5 min, 100% solution B at 9.5 min, 100% solution B at 10.5 min, and 0% solution B at 12 min. The flow rate was 0.35 mL min$^{-1}$. The autosampler was set at 5 °C, and the injection volume was 5 µL. Data were acquired in multiple reaction monitoring (MRM) mode with optimized conditions. Peak detection, integration, and quantitative analysis were performed with MassHunter Quantitative analysis software (Agilent Technologies). Blank samples were evaluated, and their injection showed no interference (no peak detected), during the analysis[65].

**Isolation of colonic mononuclear cells**. Colon was washed, cut into small pieces and resuspended three times with RPMI 10% Fetal Bovine Serum (FBS), 5 mM EDTA at 4 °C for 10 min. After washing, colon pieces were digested twice with 0.02 % collagenase VIII (Sigma-Aldrich) in RPMI, 5% FBS, 15 mM HEPES for 1 h at 37 °C. Supernatant was passed through a 70 µm cell strainer and centrifuged. Mononuclear cells were then isolated upon 30% Percoll gradient.

**Flow cytometry analysis**. Cells were suspended in PBS 10% FBS, 2 mM EDTA containing 5 µg mL$^{-1}$ anti-CD16/CD32 (clone 2.4G2; Miltenyi, Bergisch Gladbach, Germany) (blocking buffer) for 30 min at 4 °C. After washing, viability dye eFluor780 (Thermo Fisher scientific) or fixable viability stain (FVS440UV, BD Biosciences, San Diego, CA, USA) was added to the cells for 20 min at 4 °C. Cell surface antigens were then stained with optimal concentrations of fluorochrome-labeled antibodies diluted in PBS 10% FBS, 2 mM EDTA for 30 min at 4 °C (Supplementary table 3). Before transcription factor staining, cells were permeabilized using Foxp3/Transcription Factor Staining Buffer Set (eBioscience, Thermo Fisher scientific). In addition, for intracellular cytokine staining, cells were incubated with 50 ng m L$^{-1}$ phorbol myristate acetate and 500 ng m L$^{-1}$ ionomycin (Sigma-Aldrich) for 4 h. The protein transport inhibitor Brefeldin A (eBioscience) was added to the cells for the last 2 h. After washing, the cells were first incubated with blocking buffer and then with labeled anti-cytokine antibodies. Data were acquired on a Fortessa X20 and Symphony (BD Biosciences) and further analyzed using the FlowJo software (Tree Star, Ashland, OR, USA). Doublet cells and dead cells were excluded based on scatter analysis and viability dye staining respectively.

For UMAP and phenograph analysis, FCS files exported from BD FACSDiva software were imported into FlowJo software and concatenated. The FlowJo plugin UMAP (V3.1) was run on the resulting FCS file using the default settings (Distance function =

Euclidean, nearest neighbors: 15 and minimum distance: 0.5) including the following parameters: TCRβ, CD8α, TCRγδ, CD11b, FcεRI, Ly6G, Siglec F (CD170), CD103, ST2, CD117, CD4, NK1.1, MHC-II, CD11c. For cluster identification, the FlowJo plugin phenograph (V2.4) was run on the resulting UMAP using the default settings (nearest neighbors $K = 30$). Expression levels of analyzed markers were used to associate immune cell types to identify clusters according to the literature.

**Colorectal distension and electromyography recording**. Three days before colorectal distension, 2 nickel-chrome electrodes (Bioflex insulated wire AS631; Cooner Wire, Chatsworth, CA, USA) were implanted into the abdominal external oblique musculature of anaesthetized mice. Electrodes exteriorized at the back of the neck were connected via a Bio Amp (ADInstruments, Inc.) to an electromyogram acquisition system (PowerLab, ADinstruments). A 9 mm diameter balloon catheter (Fogarty catheter for arterial embolectomy, 4 F; Edwards Lifesciences, Nijmegen, Netherlands) inserted into the colon at 5 mm from the anus was progressively inflated in a stepwise of 15 mmHg. Ten-second distensions were performed at pressures of 15, 30, 45, and 60 mmHg with 5 min' rest intervals. Electromyography activity of abdominal muscles was recorded and visceromotor responses were calculated using LabChart 8 software (ADinstruments)[66].

**In vivo intestinal permeability assessment**. Mice were gavaged with a fixed concentration of either 4 kDa Fluorescein isothiocyanate (FITC)-labeled dextran (tracer of paracellular permeability) or 40 kDa FITC-dextran (tracer of transcellular permeability) (Sigma-Aldrich). Fluorescence intensity was quantified 4 h later in the serum of the animals with a fluorescence spectrometer (Insight, PerkinElmer, Waltham, Ma, USA).

**Ex vivo intestinal permeability assessment**. Biopsies from Peyer's patches, ileum and colon were mounted in Ussing chamber with an exposed tissue area of 0.196 cm$^2$ and maintained in circulating oxygenated Ringer solution at 37°C by a gas flow (95% $O_2$/5% $CO_2$). Paracellular permeability was assessed by adding 4 kDa FITC-dextran ($10^{-5}$ M) into the mucosal (apical) side and then measuring the fluorescence intensity in the serosal (basolateral) side 1 h later. Results were expressed as flux of FITC-dextran per area of mucosa (cm$^2$) per hour[67].

**Intestinal epithelial cell apoptosis assessment**. Epithelial cell apoptosis was quantified by immunochemical staining of activated caspase 3 in colonic biopsies. 5 µm colonic section were first incubated with PBS containing 3% bovine serum albumin (BSA,

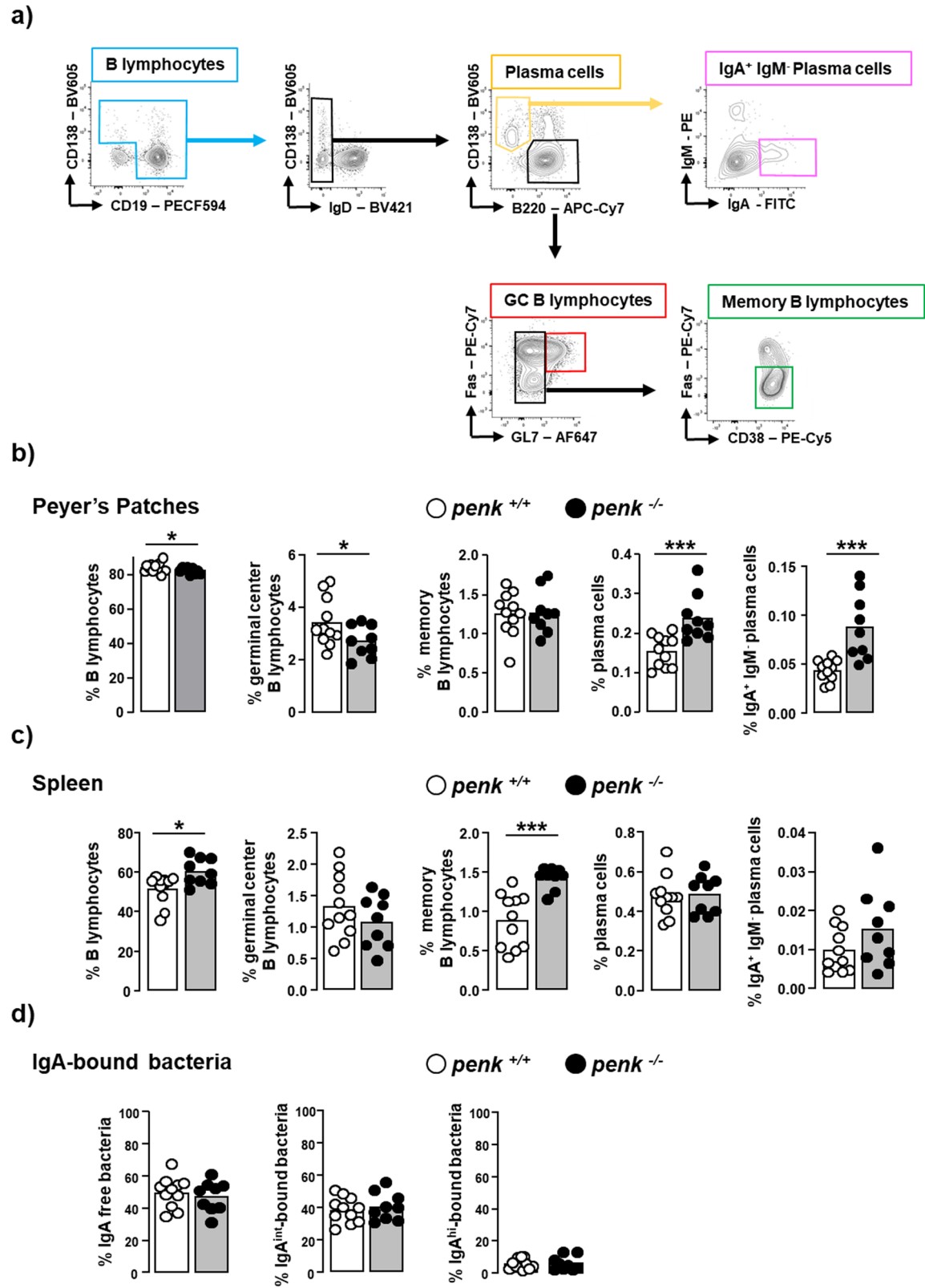

Sigma-Aldrich) and 2.5% normal donkey serum for 90 min at room temperature (RT) followed by an additional incubation with rabbit anti-activated caspase 3 polyclonal IgG antibodies (Abcam) for 90 min at RT. After extensive washing with PBS, bound antibodies were revealed by adding AlexaFluor 555-conjugated donkey anti-rabbit IgG (Life Technologies). Nuclei were then stained with DAPI for 20 min (Invitrogen) and the

slices were mounted with ProLong Gold mounting medium (Life Technologies, Carlsbad, CA, USA). Images were acquired with Zeiss 710 inverted confocal microscope (x20 NA 0.8). Apoptotic epithelial cells were quantified by measuring the activated caspase 3 density (threshold 34) normalized to the total area of the field and total number of edges (threshold 70, process binary watershed and analyze particles)[68].

**Fig. 6 Penk-deficient chimeric mice display increased IgA-expressing plasma cells in Peyer's patches.** The relative frequency of B lymphocyte subsets in Peyer's patches and spleen from penk[+/+] (white circles and white histogram) and penk[−/−] (black circles and gray histogram) chimeric mice (n = 11 versus 9) was estimated by cytofluorometry. **a** Gating strategy for flow cytometry data analysis. B cells were defined as cells expressing CD138 and/or CD19. IgD[−] IgA[+] CD138[+] B220[−] plasma cells were distinguished from B220[+] B cells. Germinal center (GC) B lymphocytes expressing the cell surface antigens Fas and GL7 were then distinguished from CD38[+] memory B lymphocytes negative for both Fas and GL7. Frequency of B lymphocytes, GC B lymphocytes, memory B lymphocytes, total plasma cells and IgA-expressing plasma cells among living cells within Peyer's patches (**b**) and spleen (**c**). **d** Frequency of non-coated fecal bacteria (left panel) and low (middle panel) and high (right panel) IgA-coated fecal bacteria. Each circle represents one mouse. Statistical analysis was performed using Mann-Whitney U test; *p < 0.05, ***p < 0.001.

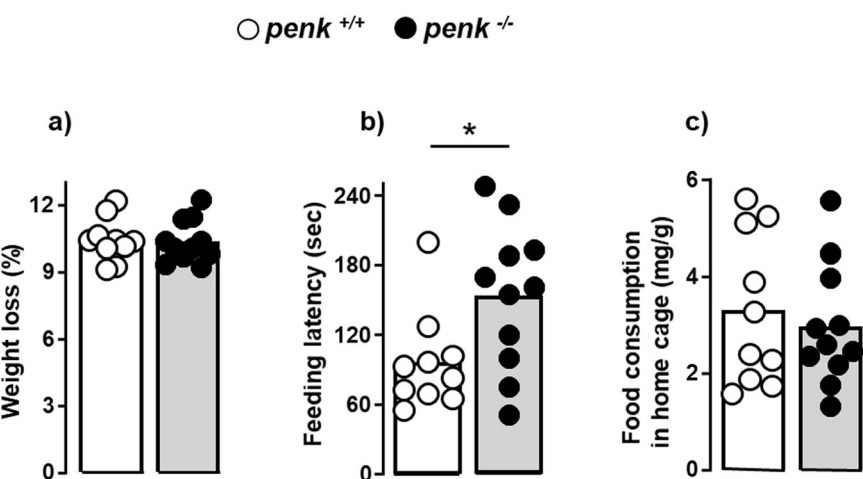

**Fig. 7 Penk-deficient chimeric mice display an anxiety-like behavior.** penk[+/+] (white circles and white histogram) and penk[−/−] (black circles and gray histogram) chimeric mice were assessed for anxiety-related behavior by using novelty-suppressed feeding test necessitating mice with similar feeding and hunger behavior (n = 10 versus 11). **a** Body weight lost induced by 24 h food deprivation prior to testing. **b** Latency time before chewing the pellet of regular food placed in the center of the brightly lit box. **c** Hunger behavior evaluated by the amount of food consumed for 5 min by the animals returned in their home cage immediately after testing. Each circle represents one mouse. Statistical analysis was performed using Mann-Whitney U test. *p < 0.05.

**Colon microbiota imaging.** Distal colon biopsies containing one faeces were fixed in Carnoy's solution prior to be paraffin-embedded. Slides were hybridized with 1 μM of a universal bacterial 16S fluorescent rRNA probe (Eub338-Cy5, 5'GCTGCCTCCCGTAGGAGT3'- Cyanine 5, Eurofins) to stain bacteria, and counterstained with both 4',6-diamidino-2-phenylindole (DAPI, Invitrogen) and fluorescein-labeled wheat germ agglutinin (ThermoFisher) to visualize host cell nuclei and sialic acids + N-acetylglucosamine content respectively. Images were acquired with a Zeiss 710 inverted confocal microscope, and FIJI freeware was used for final image mounting (v.1.51). For each animal, biofilm damage was evaluated using a scale ranging from 0 to 15 adapted with slight modifications from Motta et al. 2019 as follows: inner mucus layer invasion (rare (0), few (1), numerous (2), numerous and dense colonies (3)), biofilm distance to epithelia (>20 μm (0), >10 μm (1), numerous contact (2), dense biofilm in contact (3)), biofilm density (high (0), mild (1), scattered (2), mostly planktonic (3)), bacterial translocation into lamina propria (none (0), few (1), numerous (2), dense colonies (3)), immune cell migration into the biofilm (none (0), scattered (1), dense (2), dense and extracellular DNA (3))[23].

Mucus infiltrating-bacteria were estimated by image processing on Fiji by enumerating pixels corresponding to Cyanine 5 labeled 16S RNA between the edge of the lumen to the edge of the epithelium. The mucus area was manually traced, and an arbitrary distance was assigned to each Cyanine 5 pixel from the middle of the mucus layer to the edge of the lumen (Apical) and to the epithelial cell border (Basal). The number of pixels for each distance (strata) was determined[69].

**Taxonomic analysis of gut microbiota.** Total DNA was extracted from freshly-collected snap-frozen feces[70]. The 16 S rRNA gene V3-V4 regions were targeted by the 357wf-785R primers and analyzed by MiSeq at RTLGenomics (Texas, USA). An average of 18069 sequences per sample was generated. PCA was drawn and OTU-based diversity indices calculated with the software PAST4.10. LDA score graph was drawn by using the Huttenhower Galaxy web application via LEfSe[71].

**Assessment of IgA-coated fecal bacteria.** Feces were suspended in PBS, homogenized and centrifuged at 700 g for 5 min to remove large particles from bacteria. Non-bound immunoglobulins were then removed by centrifuging supernatant at 12,000 g for 5 min. Bacteria pellet was resuspended in PBS 1% BSA. IgA bound to fecal bacteria were stained with both FITC-labeled anti-mouse IgA heavy chain and PE-labeled anti-mouse Kappa light chain for 30 min on ice. Bacteria were then fixed with 4% PFA overnight at 4 °C. After washing, bacteria were resuspended in PBS containing 2.5 mg m L[−1] DAPI. Analysis was performed on 10,000 events at a flow rate of 1000–2000 events/s to avoid the presence of two particles in the same droplet. Each DAPI staining was considered as bacteria[72]. Data were acquired on a Fortessa X20 and further analyzed using the FlowJo software.

**Open field test.** The open field test was performed in a circular Plexiglas box (40 cm diameter) for 10 min. Experiments were performed in a room containing no conspicuous features and illuminated by a white light (30 lux). The arena was surmounted by a video camera connected to a video recorder to measure the number of entries and the time spent in the center of the open field with an automated system (Noldus, Ethovision, The Nederland).

**Elevated plus maze test**. Mice were placed in the central platform (10 × 10 cm) of a plus shaped track apparatus with two closed and two open arms (30 × 10 × 20 cm) elevated 50 cm above the floor and illuminated by a white light (25 lux). The number of entries into closed and open arms and the time spent in each arm were manually scored. Each trial lasted 5 min. The maze was cleaned with 70% ethanol between each mouse to remove olfactory cues.

**Novelty suppressed feeding test**. Feeding test paradigm elicits competing motivations between hungry up and fear to eat in the center of a brightly lit arena[73]. Briefly, after a food deprivation for 24 h, mice were weighed to control that mice exhibited a similar deprivation-induced weight loss. Mice were then placed into a corner of a box in which a pellet of regular chow was located on a white paper brightly lit in the center. The latency to start eating the food pellet was recorded. Appetite which provides a measure of hunger drive was controlled by measuring food consumption for 5 min after mice were returned to their home cage.

**Tail suspension test**. Mice, acoustically and visually isolated, were suspended 50 cm above the floor by adhesive tape placed ~1 cm from the tip of the tail. The time on which mice hung passively and completely motionless and the latency to the first immobility were recorded during a 6-min period.

**Statistics and reproducibility**. One tailed Mann-Whitney test was used to calculate statistically significant difference between the two groups of $penk^{+/+}$ and $penk^{-/-}$ chimeric mice. One-way analysis of variance (ANOVA) followed by Sidak's multiple comparison test was used to compare the means from multiple groups. Two-way ANOVA followed by Sidak's multiple comparison test was used to calculate difference significance of the visceral sensitivity between the $penk^{+/+}$ and $penk^{-/-}$ chimeric mice. $p$ values less than 0.05 were considered significant. All the statistical analyses were performed using GraphPad Prism 9.0 software (GraphPad Software, San Diego, CA). The sample sizes and number of replicates are indicated in the figure legends.

**Reporting summary**. Further information on research design is available in the Nature Portfolio Reporting Summary linked to this article.

## Data availability

The data generated in this study are provided in the main text and Supplementary Information. RNA sequencing raw data are available on the following link: https://www.ncbi.nlm.nih.gov/geo/query/acc.cgi?acc=GSE244917. Source data can be obtained from the Figshare repository with the following links: Fig. 1. https://doi.org/10.6084/m9.figshare.24466762.v1; Fig. 2. https://doi.org/10.6084/m9.figshare.24467023.v1; Fig. 3. https://doi.org/10.6084/m9.figshare.24467032.v1; Fig. 4. https://doi.org/10.6084/m9.figshare.24467275.v1; Fig. 5. https://doi.org/10.6084/m9.figshare.24467644.v1; Fig. 6. https://doi.org/10.6084/m9.figshare.24467803.v1; Fig. 7. https://doi.org/10.6084/m9.figshare.24468358.v1. All other data are available from the corresponding author (or other sources, as applicable) on reasonable request.

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

## Acknowledgements
The authors wish to thank the animal care facility, Genetoul, anexplo, US006/INSERM, Toulouse, MetaToul (Toulouse metabolomics & fluxomics facilities, www.metatoul.fr) which is part of the French National Infrastructure for Metabolomics and Fluxomics MetaboHUB-ANR-11-INBS-0010 and the platform Aninfimip, an EquipEx ('Equipement d'Excellence') supported by the French government through the Investments for the Future program (ANR-11-EQPX-0003). We gratefully acknowledge the technical assistance provided by the INFINITy (INSERM 1291) personnel of both the flow cytometry (F.E. L'Faqihi-Olive, V. Duplan-Eche and Anne-Laure Iscache) and cellular imaging (S. Allart and S. Lachambre) core facilities connected to 'Toulouse Réseau Imagerie' network. This work was supported by the Institut National de la Santé et de la Recherche Médicale (INSERM), the Université Paul Sabatier, Toulouse III, the Association François Aupetit (G.D.) and the French Agence Nationale de la Recherche (NOCICEPOP-ANR-22-CE14-0011-01, G.D.). X.M.O., L.R., N.G., C.P. and E.W. were supported by scholarship from « Ministère de l'Enseignement Supérieur, de la Recherche et de l'Innovation » and L.B. by The Fondation pour la Recherche Médicale.

## Author contributions
X.M.O., L.R., L.B., C.B., C.P., A.A., E.W., C.B., J.P.M., C.K., F.B., M.A., N.C., L.M., designed and conducted experiments. X.M.O., L.R., L.B., A.A., J.P.M., C.K., F.B., M.A., N.C., L.M., N.F., and G.D. analyzed data. E.E. and N.G. performed bioinformatics analyses from bulk RNA-sequencing, and M.S. analyzed gut microbiota taxonomy. G.D. conceived and supervised the project and wrote the manuscript. All the authors edited and approved the manuscript.

## Competing interests
The authors declare no competing interests.
