## [Peer Review File · Communications Biology]

Reviewers' comments:

Reviewer #1 (Remarks to the Author):

In this manuscript, the authors systemically analyzed the effect of enkephalin deficiency in HSC on gut immune profiles, intestinal permeability and nociception, which suggests an important role of immune cell derived enkephalin in regulating gut barrier and anxiety-like behavior at baseline. The authors generated penk chimeric bone marrow knockout mice and found that these mice displayed increased visceral hypersensitivity, intestinal permeability and altered gut microbiota. Similar to penk knockout mice, the penk chimeric knockout mice were also prone to anxiety behaviors. The study enriches our understanding of the role of enkephalins in regulating gut barrier and the relevant behavioral abnormalities.

Major comments:

1. Some of the phenotypes observed in the penk chimeric ko mice were similar to what have been described in IBS patients. However, there's no connection between penk ko and IBS pathogenesis. Could the authors provide more evidence that these phenotypes are IBS like symptoms? Or does IBS patients have lower enkephalin in the gut?

2. Fig 4 shows that Penk ko mice have increased gut permeability, which might be explained by a thinner mucus layer. Analysis of the mucus layer thickness in penk ko and control mice is strongly suggested.

Minor comments:

1. The writing of the manuscript needs to be further improved, there're grammar (e.g. Line 75-76) and typos (e.g. line 129) errors.

Reviewer #2 (Remarks to the Author):

Brief summary of the manuscript

Mas-Orea et al. reported a direct involvement of immune cells-derived proenkephalin in the regulation of basal colon sensitivity in mice. They observed that chimeric mice engrafted with enkephalin-deficient bone marrow cells develop visceral hypersensitivity, an increase in gut permeability, a damage to mucus layer with an invasion of bacteria and a higher frequency of IgA-producing plasma cells in Peyer patches. No changes in gut motility were instead observed. The authors associated this phenotype to and IBS condition.

Overall impression of the work

The topic of the manuscript is really interesting and deserve attention from the scientific community. The work is well written and sliding even if it would benefit from a revision, in particular the results should be described and discussed more in detail. The overall impression is that the description of the results is too synthetic and superficial. The discussion fails to elaborate on some aspects, such as mast cells implication and differences between the immune response in the large (where visceral sensitivity was measured) and in the small intestine (where Peyer patches are placed and where permeability can be affected too).

Specific comments, with recommendations for addressing each comment

1. The title is overstating, since it reports the enkephalin depletion results in irritable bowel syndrome-like symptoms in mice, but the authors clearly stated in the discussion that no alterations in gut motility were detected, so the effect of the genetic modification regards only some aspects associated with IBS condition.

2. In the abstract, the sentence "enkephalins play a pivotal role in the gut-brain homeostasis" is overstating too. The authors did not investigate directly the effect of enkephalin depletion in immune cells in other sites than the gut. So, they can just report an indirect effect on anxiety like behaviour, which can be attributable also to the stress related to the hypersensitive condition.

3. The description of the results should be extended.

4. It is not clear why the authors chose the novelty-suppressed feeding test to assess anxiety in

mice. Please, clarify. Did the animals display also depressive like behaviours?

5. Figure 1. The graphs should be rescaled and uniformed to adequately show the results. Images in "c" seems to be out of the context. Were a quantification of immunofluorescence signal quantified? A clarification about figure 1e and 1f results is needed.

6. Figure 2. The percentage of mast cells seems to be significantly affected in *penk*^{-/-} mice. How do the authors explain this phenomenon and how can it be linked with visceral sensitivity increase?

7. Figure 4 and 5. It is not clear when the assessment of visceral sensitivity was performed in the animals, temporally. Did the authors perform a time course analysis after the chimeric procedure? The interpretation of the data would highly benefit from an experimental scheme showing each step and where each analysis has been performed (colon or small intestine). The authors should also clarify why the assessed permeability in the small intestine and biofilm studies in the colon where sensitivity was actually measured.

8. Discussion. "The reduction of the enkephalin tone in basal situations also resulted in an increase in both epithelial paracellular and transcellular permeability together with a spatial redistribution of the luminal bacteria". How can this phenomenon be explained?

9. Discussion. The authors pass from the description of the colon to the small changes (Peyer's patches... colitis... microbiota invasion...) without following a biological thread. It makes difficult to drive conclusions and to create a story.

10. Discussion. Does the antagonism on enkephalin targets promote the same phenotype observed in *penk*^{-/-} mice?

11. Discussion. Enkephalins are also involved in intestinal secretory activity, but the authors seem not to take in account this aspect.

First of all, we would like to thank the two referees for the evaluation of our work, attentive reading of our manuscript and their constructive comments.

Reviewer #1:

Major comments:

1. Some of the phenotypes observed in the penk chimeric ko mice were similar to what have been described in IBS patients but there's no direct connection between penk ko and IBS pathogenesis.

We agree with the two reviewers that enkephalin deficiency in immune cells reproduces only some symptoms of IBS and, particularly, it does not alter gut motility which is a main hallmark of IBS. So, since we have not, at time, a direct connection between enkephalin deficiency and IBS pathogenesis, we changed the title of the manuscript as "Proenkephalin deletion in hematopoietic cells induces intestinal barrier failure resulting in clinical feature similarities with irritable bowel syndrome in mice".

Could the authors provide more evidence that IBS patients have lower enkephalin in the gut?

In line with the prevailing, if not exclusive, production of enkephalins by mucosal immune cells^{1,2}, proopiomelanocortin mRNA coding β -endorphin is virtually undetectable in mouse colonic mucosa^{3,4}. This is different in humans, as mucosal immune cells predominantly macrophages and, to a lesser extent, T lymphocytes express β -endorphin. In agreement with a loss of endogenous regulation of colonic sensory afferents by immune-derived opioids in IBS, β -endorphin expression was found reduced in colonic biopsies from IBS patients as compared to healthy controls⁵. This point is now discussed (Discussion section: Page 13, second paragraph). Two new references have been added.

2. Fig 4 shows that Penk ko mice have increased gut permeability, which might be explained by a thinner mucus layer. Analysis of the mucus layer thickness in penk ko and control mice is strongly suggested.

Measurement of the mucus layer thickness has been done as requested by the referee. The results indicating no difference between the two groups of mice are depicted in the new figure 5 panel c. The results are described (page 8, line 6) and discussed (page 12, lines 13 - 15). A new reference has been added.

Minor comments:

1. The writing of the manuscript needs to be further improved, there're grammar (e.g. Line 75-76) and typos (e.g. line 129) errors

Grammar and typos errors, lines 75-76 and line 129 respectively have been corrected.

Reviewer #2:

1. The title is overstating, since it reports the enkephalin depletion results in irritable bowel syndrome-like symptoms in mice, but the authors clearly stated in the discussion that no alterations in gut motility were detected, so the effect of the genetic modification regards only some aspects associated with IBS condition.

We agree with the two reviewers that enkephalin deficiency in immune cells reproduces only some symptoms of IBS and, particularly, it does not alter gut motility which is a main hallmark of IBS. So, since we have not, at time, a direct connection between enkephalin deficiency and IBS pathogenesis, we changed the title of the manuscript as “Proenkephalin deletion in hematopoietic cells induces intestinal barrier failure resulting in clinical feature similarities with irritable bowel syndrome in mice”.

2. In the abstract, the sentence “enkephalins play a pivotal role in the gut-brain homeostasis” is overstating too. The authors did not investigate directly the effect of enkephalin depletion in immune cells in other sites than the gut. So, they can just report an indirect effect on anxiety like behavior, which can be attributable also to the stress related to the hypersensitive condition.

We believe, like our referee, that anxiety-like behavior is rather attributable to the stress related to the hypersensitive condition than a direct effect in the brain. So as requested by the referee, the sentence has been modified as follows: “Thus, our results show that immune cell-derived enkephalins play a pivotal role in maintaining gut homeostasis and normal behavior in mice”.

3. The description of the results should be extended.

The description of the results has been improved to be less synthetic as requested.

4. It is not clear why the authors chose the novelty-suppressed feeding test to assess anxiety in mice. Please, clarify. Did the animals display also depressive like behaviours?

Mice were assessed for anxiety using open field, elevated plus-maze and the novelty suppressed feeding tests. They were also assessed for depression-like behavior using tail suspension test (all the protocols are now added to the Materials and Methods section). Only

the novelty suppressed feeding pointed out anxiety-like behavior in $Penk^{-/-}$ chimeras probably because of the higher anxiogenic properties of this test. The results are now depicted in the supplementary figure 7 and described in the relevant section.

5. Figure 1. The graphs should be rescaled and uniformed to adequately show the results. Images in “d” seems to be out of the context. Were a quantification of immunofluorescence signal quantified?

The graphs in Figure 1 have been rescaled and uniformed. We agree with the reviewer that the images in “d” are out of the context since at this point of the manuscript there is no indication about $penk$ in T cells. In addition, in the absence of a quantification of immunofluorescence signals, we cannot formally justify the absence of T lymphocytes, the main hematopoietic source of enkephalins, in colonic tissue areas involved in gut motility. Thus, the images have been removed from the figure 1.

A clarification about figure 1e and 1f results is needed.

Figure 1e and 1f (panels 1d and 1e in the new figure 1) results as well as the corresponding figure legends are now clarified.

6. Figure 2. The percentage of mast cells seems to be significantly affected in $penk^{-/-}$ mice. How do the authors explain this phenomenon and how can it be linked with visceral sensitivity increase?

As mentioned by the referee, the relative number of mast cells among $CD45^{+}$ hematopoietic cells, was significantly reduced in $penk^{-/-}$ mice. However, the density of mast cells per mm^2 of colonic mucosa remained unchanged (Figure below). Given the low percentage of mast cells, their relative decrease (around 0.05%) may be mostly because of the increase in $CD4^{+}$ T lymphocytes. In biopsies of IBS patients mast cells have been often shown to be in closer proximity to colonic nerve endings, a finding that was correlated to abdominal pain intensity. Our experimental design did not allow to argue for a similar mechanism in $penk^{-/-}$ chimeric mice.

Mucosal colonic mast cell density is not altered in Penk-deficient chimeric mice.

Lethally irradiated C57BL/6JRj wild-type mice were engrafted with bone marrow cells from either penk^{+/+} (white circles and white histogram) and penk^{-/-} mice (black circles and grey histogram). Eighteen weeks after, chimeric mice were examined for density of both mast cells and CD3⁺ T lymphocytes within colonic lamina propria. Colonic sections saturated with PBS 1% FCS were incubated with either rabbit anti-CD3 (Clone SP7, Diagnostic BioSystems, Pleasanton, CA, USA) or rat anti- mucosal mast cell protease (mMCP)-1/MCPT1 (Clone RF6.1, Invitrogen, Waltham, MA, USA) monoclonal antibodies (mAb) for 1 hour at room temperature. After washing, bound antibodies were revealed with Alexa Fluor 555-labeled donkey anti-rabbit IgG antibodies (Invitrogen) or Alexa Fluor™ Plus 488-labeled goat anti-rat IgG antibodies (Thermo Fisher scientific, Waltham, MA, USA) respectively. Slides were mounted and nuclei were stained with 4',6-Diamidino-2-Phenylindole (DAPI). Images were taken using a Zeiss 710 inverted confocal microscope (Carl Zeiss microscopy GmbH, Jena, Germany) with x 20 objective. Mast cells were quantified per mm² of tissue. Each point, corresponding to one animal, represents the mean of more than four different histological examinations per slice. Statistical analysis was performed using Mann-Whitney U test. No significant difference was observed. A representative immunostaining (scale 50 μm) is shown.

7. Figure 4 and 5. It is not clear when the assessment of visceral sensitivity was performed in the animals, temporally. Did the authors perform a time course analysis after the chimeric procedure? The interpretation of the data would highly benefit from an experimental scheme showing each step and where each analysis has been performed (colon or small intestine).

The assessment of visceral sensitivity was performed only at one time-point. No time course analysis was done. A schematic representation of the experimental design indicating where each analysis has been performed (colon or small intestine) is now included as supplementary table 1.

The authors should also clarify why the assessed permeability in the small intestine and biofilm studies in the colon where sensitivity was actually measured.

The permeability was assessed in the small intestine to show that the increase in intestinal permeability observed *in vivo* was not restricted to the colon where the sensitivity was actually measured. Furthermore, the increased permeability in the small intestine justified the analysis of IgA-producing plasma cells in Peyer's patches to look at whether the increased traffic of luminal antigens across the intestinal epithelium induced a T cell-dependent IgA antibody response.

Why gut microbiota organization analysis has been performed on distal colon but not on the small intestine?

In the distal colon, the epithelium is covered by a dense mucus layer ('inner layer') free of bacteria and, above it, a loosely adherent mucus layer ('outer layer') housing most of the bacteria living in dense communities (biofilms). In healthy conditions, local adherent microbial communities (biofilms) resides on the surface of the inner mucus layer and typically don't go beyond this layer. In some pathological conditions including low-grade inflammation, the density and/or morphotypes of bacteria, their ability to penetrate the inner mucus layer, to interact with epithelium or to translocate into *lamina propria* can change ⁶. In the upper part of the intestine, like the ileum, the spatial organization is quite different. The mucus layer no longer lines the surface of the mucosa, except in deeper crypts, and instead forms loosely attached clusters on top of the villi. The arrangement of microorganisms there tends to be more clustered, either on the surface of the epithelial cells or attached to leftover food particles. There are more frequent interactions between the epithelial cells and bacteria such as segmented filamentous bacteria. Hence, unless there is severe inflammation with clear signs of bacteria disorganization, or moving through the tissue, looking at the microbial organization on the ileum makes it challenging to distinguish between a healthy state and a slightly inflamed condition, as we anticipated in our study ⁷.

from ⁷

This point is now clarified in the results section (page 7, last paragraph).

8. Discussion. “The reduction of the enkephalin tone in basal situations also resulted in an increase in both epithelial paracellular and transcellular permeability together with a spatial redistribution of the luminal bacteria”. How can this phenomenon be explained?

The host - microbiota relationship is bidirectional. A shift in spatial organization of microbes can lead to increased proximity with the epithelium. This increased proximity of microbiota with epithelium may then induce, via TLRs, the activation of NFκB, a master transcriptional regulator well known for its role in heightening paracellular permeability as exemplified by the TNF-induced tight junction barrier alteration ⁸. In turn, intestinal inflammation can impact the equilibrium of mucosa-associated microbial communities and shift the spatial organization of commensal microbes ⁹. Nociceptor hyperactivity associated to the increased visceral sensitivity in enkephalin-deficient mice may also potentially elicit abnormalities in protein components of junctional complexes regulating epithelial cell interactions.

9. Discussion. The authors pass from the description of the colon to the small intestine changes (Peyer’s patches... colitis... microbiota invasion...) without following a biological thread. It makes difficult to drive conclusions and to create a story.

This point has been improved in the new version of the manuscript. We hope this will satisfy referee expectation. However, the pleiotropic effects of opioids on physiology make hard to draw a satisfactory biological thread.

10. Discussion. Does the antagonism on enkephalin targets promote the same phenotype observed in *penk*^{-/-} mice?

This point is now discussed (page 14, lines 1-3). It is illustrated with a new supplementary figure 9 showing that chronic treatment with naloxone-methiodide of wild-type C57BL/6 mice in basal conditions does not affect intestinal permeability.

11. Discussion. Enkephalins are also involved in intestinal secretory activity, but the authors seem not to take in account this aspect.

Opioids are known to inhibit secretion of both fluid and electrolyte in intestine but unfortunately, we were not able to measure it. The absence of any alterations in both the consistency and frequency of the stools is however indicative for the innocuity of immune-derived enkephalins on the intestinal fluid secretion.

The effect of enkephalin deficiency in immune cells on the levels of mRNA coding for a number of protein components of mucus and antimicrobial peptides secreted by intestinal epithelial cells are now reported in the supplementary figure 8 and discussed. The primers are listed in the supplementary table 2.

In addition, we added the expression levels of mRNAs coding a number of mediators involved in epithelium repair/regeneration in the same supplementary figure 8. The results are reported in the discussion section. The primers are listed in the supplementary table 2.

- 1 Basso, L. *et al.* T-lymphocyte-derived enkephalins reduce Th1/Th17 colitis and associated pain in mice. *J Gastroenterol* **53**, 215-226, doi:10.1007/s00535-017-1341-2 (2018).
- 2 Boue, J. *et al.* Endogenous regulation of visceral pain via production of opioids by colitogenic CD4(+) T cells in mice. *Gastroenterology* **146**, 166-175, doi:10.1053/j.gastro.2013.09.020 (2014).
- 3 Czarnewski, P. *et al.* Conserved transcriptomic profile between mouse and human colitis allows unsupervised patient stratification. *Nat Commun* **10**, 2892, doi:10.1038/s41467-019-10769-x (2019).
- 4 Reiss, D. *et al.* Mu and delta opioid receptor knockout mice show increased colonic sensitivity. *Eur J Pain* **21**, 623-634, doi:10.1002/ejp.965 (2017).
- 5 Hughes, P. A. *et al.* Immune derived opioidergic inhibition of viscerosensory afferents is decreased in Irritable Bowel Syndrome patients. *Brain Behav Immun* **42**, 191-203, doi:10.1016/j.bbi.2014.07.001 (2014).
- 6 Motta, J. P. *et al.* Active thrombin produced by the intestinal epithelium controls mucosal biofilms. *Nat Commun* **10**, 3224, doi:10.1038/s41467-019-11140-w (2019).
- 7 Motta, J. P., Wallace, J. L., Buret, A. G., Deraison, C. & Vergnolle, N. Gastrointestinal biofilms in health and disease. *Nat Rev Gastroenterol Hepatol* **18**, 314-334, doi:10.1038/s41575-020-00397-y (2021).
- 8 Wang, F. *et al.* Interferon-gamma and tumor necrosis factor-alpha synergize to induce intestinal epithelial barrier dysfunction by up-regulating myosin light chain kinase expression. *Am J Pathol* **166**, 409-419, doi:10.1016/s0002-9440(10)62264-x (2005).

- 9 Chanin, R. B. *et al.* Epithelial-Derived Reactive Oxygen Species Enable AppBCX-Mediated Aerobic Respiration of Escherichia coli during Intestinal Inflammation. *Cell Host Microbe* **28**, 780-788 e785, doi:10.1016/j.chom.2020.09.005 (2020).

REVIEWERS' COMMENTS:

Reviewer #1 (Remarks to the Author):

All the comments have been well addressed. I congratulate the authors on this nice study.

Reviewer #2 (Remarks to the Author):

I appreciated the Revision of the manuscript provided by the authors. Therefore, considering the results of the work really interesting, I agree with the acceptance of the manuscript for publishing in Communications biology.